# Evaluation of Psychophysical Fitness in Drivers over 65 Years of Age

**DOI:** 10.3390/healthcare11131927

**Published:** 2023-07-03

**Authors:** Enrique Mirabet, Macarena Tortosa-Perez, Francisco Tortosa, Francisco González-Sala

**Affiliations:** 1Instituto de Investigación en Tráfico y Seguridad Vial-INTRAS, Universidad de Valencia, 46010 Valencia, Spain; emirabet@comv.es (E.M.); francisco.m.tortosa@uv.es (F.T.); 2Departamento de Psicología, Universidad Internacional de Valencia-VIU, 46002 Valencia, Spain; macarena.tortosap@campusviu.es; 3Departamento de Psicología Evolutiva y de la Educación, Universidad de Valencia, 46010 Valencia, Spain

**Keywords:** older adult, psychophysical aptitudes, assessment aptitudes, older drivers, road safety

## Abstract

Background: The deterioration of cognitive and psychophysical ability associated with aging has an effect on road safety, especially in the driving of vehicles. The current study’s main objective is to evaluate the psychophysical aptitudes in drivers over 65 years of age in a sample of drivers in Spain. Methods: The sample was formed of a total of 1663 drivers who attended a Driver Recognition Center. The evaluation of their psychophysical aptitudes was carried out following the Medical-Psychological Exploration Protocol for Driver Recognition Centers, edited by the Ministry of Health and the General Directorate of Traffic. Results: The results show increased restrictions in the evaluation of driving ability with age, which are especially significant after 75 years of age. Regarding sex, 70.1% of women have an approved evaluation, compared to men aged between 65–69, although from 69 onwards, the percentage of approved women decreases significantly. The loss of visual capabilities and poor performance in psycho-technical tests are the main causes associated with an evaluation with restrictions, with the number of restrictive conditions increasing with age. Conclusions: There is an increase in the number of cases with age-related restrictions, especially in the case of women and ophthalmologic-related problems, although the majority of drivers over 65 years old continue driving, thus continuing with a practice that has been related to the well-being and quality of life of older adults.

## 1. Introduction

Aging is associated with the natural decline of neurophysiological and psychological resources [1,2], which favors an increase in the number and severity of many cognitive, motor and sensory limitations and alterations, affecting the decision-making process in ambiguous, complex and changing situations [3,4]. These situations are present in driving, thus there is an increase in the risk of accidents in older drivers [5,6,7]. These changes, not always pathological, can affect various domains, some of which are closely related to daily activities, including driving [8,9]. As a consequence, the quality of many of the skills should be measured with reliable instruments for vulnerable people [10,11]. These skills are also necessary as safe driving performance decreases [12,13,14,15,16,17,18], which is reflected in crash rates and their severity. The involvement of older people in fatal accidents is estimated to increase by 155% by 2030, accounting for 54% of the total projected increase in the number of total fatal accidents [19].

Driving is a highly complex cognitive-motor task, requiring continuous information integration from multiple sensory modalities, cognitive processes and motor actions [20,21,22]. For this reason, different countries have proposed different modalities, criteria and instruments for access to driving licenses and permits [23,24,25,26]. In Spain, this is in the Driver Recognition Centers authorized by the General Directorate of Traffic (DGT), where it is possible to renew a driving license and to obtain certificates of the physical, mental and coordination aptitude necessary to obtain or renew a driving license and authorizations for any type of vehicle [27]. In these centers, psychological, visual, auditory and general medical tests are carried out. If necessary, specific tests can also be carried out, following a recently updated protocol [28].

In Spain, current regulations establish that driver’s license renewal for people under 65 years of age is performed every 10 years in the case of driving licenses for cars, motorcycles and mopeds and every 5 years in the case of licenses for heavier vehicles (trucks, buses). In the case of people over 65 years of age, the validity periods are reduced to 5 and 3 years, respectively. However, in each of the examinations to obtain or extend the validity of these licenses, as a result of the evaluation, a shorter validity than normal may be imposed, up to one year or less, due to psychophysical causes defined in Annex IV of the Royal Decree 818/2009 [29].

In order to analyze the road risk posed by the group of drivers over 65 years of age, data from the DGT’s statistical yearbook [30] show that the group of 75 years of age and above is the one with the highest number of drivers involved with accident victims. This is mainly when driving cars on urban and interurban roads, with a total of 3114 drivers involved, compared to 2450 drivers between 69 and 74 years of age and 3114 drivers between 65 and 69 years of age. This same trend by age is also maintained in relation to the number of drivers killed in the case of passenger cars without trailers on these two types of roads, with the 75 and over age group being the most represented on interurban roads, with 47 victims more than other age groups, while on urban roads, it is the most prominent with 8 fatalities together with the 35 to 39 age group. On the other hand, vulnerability associated with age must be qualified. So, it is to be expected that older people are more vulnerable to the consequences of a traffic accident, thus increasing the probability of death compared to younger drivers. In this sense, when addressing the issue of drivers over 65 years of age, not only should whether they pose a greater risk to traffic in general be taken into account but also the risk they pose to themselves given the fragility in health associated with age.

This situation would seem to make it advisable to establish very drastic limitations on the freedom to drive for those who make up this group, which is growing in number at a very fast pace. However, strategies that advocate for necessary healthy aging, often closely linked to the use of vehicles, especially in rural environments, call for caution in the adoption of this type of aversive measure of a legal nature, and therefore, of universal scope [31,32,33]. It should be remembered that the right to free mobility, adapted to the needs of individuals (inclusive), is a universal right, which could go back as far as Article 13 of the Universal Declaration of Human Rights [34] or the European Community Directive 2004/38 [35].

In Spain, it is the centers who determine whether permits do not have any type of restriction or whether there are adaptations, restrictions or limitations on people, vehicles or the conditions of circulation. Although the literature has abundant cross-sectional studies on large samples of drivers evaluated in these centers, showing a higher number of incidences in people over 65 years of age than in minors [36,37,38], there are not as many longitudinal studies that tackle the deterioration of psychophysical aptitudes in older drivers [39].

The sex of drivers is also related to different factors associated with driving. The authors of [40] point out that older women usually take more measures when limiting their driving, such as driving at night or on motorways, than men do. The authors of [41] also highlight differences between men and women in self-regulation when driving. Similarly, [42] suggest that, voluntarily, women usually renounce driving to a greater extent than men do for health reasons. These authors find differences according to sex in relation to the type of illness associated with a restrictive driving measure. While musculoskeletal, respiratory, ophthalmological, psychological and metabolic illnesses are predominant in women, cardiovascular illnesses and cancer are most prominent among men. The authors of [43] find that women, despite have greater visual ability than men, restrict their driving more. In particular, for women, low-contrast acuteness in glares is the variable that most highly predicts their self-limitation when driving, while for men it is the sensibility of contrast.

In the case of Spanish regulation, as is explicit in the Royal Decree 818/2009, psychophysical aptitude tests aim to evaluate the ability of an individual to drive a vehicle. These tests evaluate factors related to visual and auditory abilities; the locomotor system; cardiovascular, kidney, nervous and muscular systems; the presence of metabolic or endocrine illnesses; and the presence of psychological disorders, among others, that can impair one’s ability to drive.

The current study’s main objective is to evaluate psychophysical aptitudes, such as motor, psychological and sensory abilities, and the illnesses and treatments that lead to “approved”, “approved with restrictions” or “not fit” evaluations in drivers over 65 years old that come to a Driver Recognition Center to renew their driver’s license. As a main hypothesis of the study, it is predicted that the number of drivers that continue driving after an evaluation of their psychophysical aptitudes will be high if the number of drivers with an “approved” or “approved with restrictions” evaluation is taken into account.

Besides the current hypothesis, the following specific hypotheses have been developed:

**Hypothesis 1.** 
*There will be a significant reduction with age in the number of drivers with an approved evaluation, thus increasing the number of drivers with an “approved with restrictions” evaluation. Furthermore, it is expected that these differences will be greater in the group of drivers aged 75 or over, in comparison to other age groups.*


**Hypothesis 2.** 
*It is expected that statistically significant sex differences in the number of drivers according to the type of evaluation will be seen. In this sense, it is predicted that there will be a greater number of women with an approved evaluation compared to men. Although, taking into account age groups, the reduction in the number of approved drivers is greater in the case of women compared to men.*


**Hypothesis 3.** 
*There will be problems associated with a decline in visual ability while driving and not passing psychophysical tests in the most frequent factors when obtaining an approved with restrictions, interrupted or unfit evaluation.*


**Hypothesis 4.** 
*It is expected that the number of conditions leading to an interrupted or unfit evaluation will increase with age.*


## 2. Methods

### 2.1. Sample

Information was collected from 1663 drivers over the age of 65 who went to a Driver Recognition Center (CRC V-0001) to be evaluated to extend their driving license. Of these, 1288 (77.4%) were men and 375 (22.6%) were women. Table 1 shows the distribution by age group and sex of the drivers who made up the total sample.

### 2.2. Procedure

In accordance with the EU Directive and the General Regulations for Drivers, the cut-off point for driving license validity periods is 65 years of age. Likewise, in the General Directorate of Traffic’s Statistical Yearbook, there are three groups of older drivers (65–69; 70–74; >75), which is the reason for dividing the study sample into these three age groups.

The information was collected by the investigators and compiled in a database (Excel) in which the most significant variables from the Clinical History (Medical-Psychological Examination Protocol for Driver Examination Centers edited by the Ministry of Health and the GDT) were included. The variables extracted from the clinical history were age, sex (men/women), result (approved/approved with restriction/interrupted/unfit), restrictive diseases (ANNEX IV of the RGC, referring to diseases and deficiencies that would be cause for refusal or adaptations, driving restrictions and other limitations in obtaining or extending a driving license or permit) and, finally, restrictive conditions. In all cases, participants were notified of the purpose of the study and informed consent was sought. This study was approved by the Research Ethics Committee of the University Research Institute on Traffic and Road Safety at the University of Valencia with the reference number E0004121121.

All this information was extracted through the GICeWIN management program, which collects the variables included and coded in the Clinical History. Thus, the diseases are coded according to the application developed by the DGT and the codes of the restrictive conditions according to Section B of ANNEX I, Community driving license, of the General Rules of Driving (RD818/09). In said rules, two categories are established: restrictive disease, which refers to a disease that implies some kind of restriction (Annex IV of the GCR), and restriction, measures that imply limitation, restriction or adaptation of driving (codes in Annex I of the GCR).

The evaluation of drivers at the Driver Recognition Center can result in four possible outcomes of psychophysical evaluation. Approved: there is no evidence of diseases or deficiencies that may be cause for denial or adaptations, circulation restrictions or other limitations in the extension of a driving license. Approved with restriction: there is evidence of diseases or deficiencies that may be cause for adaptations, driving restrictions and other limitations in the extension of the driving license. The need to drive with glasses or contact lenses is not considered a restrictive condition. Interrupted: there is evidence of diseases or deficiencies that may be cause for denial or adaptations, traffic restrictions and other limitations in the extension of a driving license. It is necessary to provide a report, repeat deferred tests and/or comply with a prescribed period of time in certain medical, therapeutic or surgical conditions. Unfit: there is evidence of diseases or deficiencies that are cause for refusal or extension of a driving license.

### 2.3. Analysis

Frequency counts and calculation of percentages were performed. Chi-square tests [44] were used to determine the existence of statistically significant differences according to the type of condition, sex and age group with a significance level of *p* < 0.05. All this was carried out using the SPSS 25 statistical package.

## 3. Results

### 3.1. Number and Percentage of Drivers by Condition of the Evaluation, Regarding Age Group and Gender

As can be seen in Table 2, 47.5% of the drivers obtained a rating of approved, while 49.8% received an evaluation of approved with restrictions. According to sex, there is a higher percentage of women (61.9%) who were approved compared to men, while in the approved with restrictions category, there is a higher presence of men (53.7%). The percentages in the categories of interrupted and unfit do not exceed 2% in both men and women. These differences turned out to be statistically significant (χ^2^ = 39.596, 3df, *p* < 0.001, Φ = 0.154).

When analyzing the percentage of cases by age group according to the result of the examination (see Table 2), we observed a decrease in the number of approved cases with age, from 72% in the group between 65 and 69 years of age to 13.2% in the group over 75 years of age. On the other hand, there was an increase in the number of approved drivers with restrictions with age and in the number of drivers with an evaluation of interrupted and unfit, which reached 5.7% in the group over 75 years of age. These differences were found to be statistically significant (χ^2^ = 353.676, 6df, *p* = 0.001, Φ = 0.461).

According to sex and age group, there were statistically significant differences (χ^2^ = 96.640, 2df, *p* < 0.001, Φ = 0.241) among drivers. As shown in Table 1, there is a higher percentage of men in all age groups, with these differences being greater as age increases.

Table 3 shows the percentages by condition, sex and age group. When these results are analyzed, it can be seen in the approved condition that, in the case of men, 47.9% belong to the group between 65 and 69 years of age, which decreased to 8.8% in the case of the group over 75 years of age. It is observed that the greatest reduction occurs after 74 years of age. In relation to able-bodied drivers with restrictions, there was an increase of 28.9 percentage points with age, with these percentages being higher in the case of men with an evaluation of interrupted and unfit, reaching approximately 69 percentage points in the latter case. These differences turned out to be statistically significant (χ^2^ = 267.011, 6df, *p* = 0.001, Φ = 0.455).

In the case of women, for an evaluation of approved, there is a decrease of approximately 66 percentage points with age, being much smaller in relation to an evaluation of approved with restrictions. These differences were found to be statistically significant (χ^2^ = 65.752, 6df, *p* < 0.001, Φ = 0.419).

This trend is maintained when analyzing the total data, with a decrease in the number of approved drivers due to age and an increase in the percentages in the categories of approved with restrictions (approximately 22%), interrupted (approximately 40%) and unfit (approximately 59%) as age increases. These differences were found to be statistically significant (χ^2^ = 354.565, 6df, *p* < 0.001, Φ = 0.462).

### 3.2. Compensatory Measures for Driving: Optical Correction Measures

When examining the existence of statistical differences between approved and approved with restrictions in relation to the use of optical correction measures (see Table 4), there are no statistically significant differences between both groups (χ^2^ = 0.391, 1df, *p* = 0.532, Φ = 0.016), with approximately 70% of drivers requiring these measures in both groups. With respect to sex, there were statistically significant differences between men and women (χ^2^ = 11.720, 1df, *p* = 0.001, Φ = −0.085), with 71.4% of men requiring this type of correction compared to 62.1% of women. In regard to the age group, these differences were also statistically significant (χ^2^ = 9.863, 2df, *p* = 0.007, Φ = 0.075), with 64.6% of drivers between 65 and 69 years of age requiring optical corrections compared to 71.5% of drivers between 70 and 74 years of age and 72.7% of drivers over 75 years of age.

In relation to sex in the approved group (see Table 4), the percentages of drivers requiring optical correction are very similar to those in the group aged 65 to 69 years, the differences being greater in the case of women (75%) compared to men (68.9%). However, this trend is reversed in the group aged 75 to 79 years, with a greater presence of men (79.2%) than women (44.4%). In the group of approved with restrictions, it is men in all age groups who require optical corrections to a greater extent than women, with these differences being greater in the 65 to 69 age group.

### 3.3. Psychophysical Fitness

When analyzing the causes that led to an evaluation of approved with restrictions, it can be observed that in 74.8% of the cases, there was only one cause as a restriction. Among these causes, the most frequent were surgical aphakia (n = 309, 27.8%), progressive deterioration of visual capacity (n = 206, 18.5%), visual-motor coordination (n = 114, 10.3%) and visual acuity (n = 108, 9.7%). Visual ability problems as a cause of restriction accounted for 51.3% of the causes (n = 570), if the number of cases in the first five diseases listed in Table 5 is taken into account.

Failure to pass the psycho-technical tests, which include estimations of movement and visual-motor coordination, accounted for 16.1%. Impaired hearing ability (hearing acuity) was a condition in 9.7%. A similar percentage (8.6%) corresponded to cardiovascular disorders. SAHS (sleep apnea-hypopnea syndrome) alone accounted for 3.1% of the restrictive causes. Finally, diabetes was considered to be an influential cause in 7.8%, although it was present in a much higher percentage but it was not considered to have an influence in all cases.

In the case of drivers with restrictions, 83.3% of drivers were prescribed one restriction, 13.8% two restrictions and 2.9% three or more. The restrictions in force accounted for 77.4%, 16.3% for vehicle adaptations and 6.3% for driving restrictions. Of the periods of validity, 3 years accounted for 46.6% of all the periods of validity imposed. All these data can be consulted in Table 6.

When analyzing the most frequent restrictions, such as those associated with the period of validity, the percentages of drivers are very similar in the three age groups, ranging from 76% in the group over 75 years of age to 78.9% in the group aged 70–74. However, a higher percentage of the 3-year validity period is observed in the 65–69 age group (71.8%) compared to 29.3% in the group over 75 years of age. The inverse trend is observed when the period is for one year, with 4.7% in the 65–69 age group and 23% in the older age group.

When analyzing the psychophysical conditions that gave rise to an evaluation of unfit or interrupted (see Table 7), it can be seen that among the most frequent causes are a lack of visuomotor coordination that makes driving impossible (22.1%) and binocular visual acuity below 0.5 (16.2%). Among the different psychophysical conditions, those affecting visual capacity (30.4%) and those resulting from poor performance of psycho-technical tests (34.8%) are the main causes of refusal. Then, there are approximate CNS disorders (16%), where cerebrovascular accidents are included. A somewhat lower percentage is presented by mental disorders (10.4%), where cognitive impairments and mood alterations are the most representative mental state conditions. If the age variable is taken into account, it can be seen that the number of drivers with some restrictive conditions increases.

When analyzing the number of conditions leading to an evaluation of approved with restrictions in the group of drivers, it can be seen that this increase with age, going from 90.2% in the 65–69 age group in the case of a single condition to 63.8% in the older group, increases by approximately 21 percentage points according to age in the case of two conditions leading to an evaluation with restrictions. The rest of the results can be found in Table 8.

## 4. Discussion

The increase in the population over 65 years of age is evident if we take into account the inversion of the population pyramid in Spain and Europe, mainly increasing, according to the European Union, to form 30% of the population by 2050. Undoubtedly, this will also mean an increase in the number of elderly people as drivers, passengers, cyclists or pedestrians occupying public roads to move from one place to another, being, therefore, subjects directly or indirectly involved in driving. This mobility is a quality of well-being, social integration and independence, all indicators related to healthy aging, and is associated with a greater well-being and a better quality of life [45].

In relation to the drivers who were evaluated at the driver evaluation center, the results show that practically the majority of drivers over 65 years of age (97.3%) obtain an evaluation that allows them to continue driving with or without restrictions. The pass rate is 50% of the drivers tested, which confirms the hypothesis at the beginning of this study. These results are consistent with those of other similar studies [46].

Age is another determining variable in relation to the evaluation obtained by drivers, with the number of cases with restrictions increasing as age increases, being especially relevant after 75 years of age, thus fulfilling the first of the study’s hypotheses. This is related to the decrease of approximately 60 percentage points in the number of approved drivers between the 65 to 69 age group and the 75 and over age group and a significant increase in the percentage of cases with an evaluation of interrupted or unfit. However, the percentage with an evaluation of interrupted or unfit does not exceed 6% in the older age group. This decrease is evident in both men and women. This decline in the performance of driving as well as the cognitive abilities associated with the increase in age has been pointed out in different studies [21,47,48]; although, the big differences between individuals that can be seen in this case must be taken into consideration [20,49]. In this regard, the evaluation of psychophysical abilities in drivers becomes more relevant than chronological age, and these evaluations are more precise at the time of predicting driving performance [20,50].

In this sense, there is a higher percentage of women than men with an approved evaluation. This trend is reversed when the assessment is passed with restrictions, thus fulfilling the second hypothesis of the study. This may be the result of a sex gap with respect to driving that has been dragging on for years and/or that women stop driving earlier than men, motivated either by factors associated with a lower perception of safety as age increases or due to a certain attitudinal component associated with sex roles. Furthermore, a change in the trends regarding the number of men and women with an approved or approved with restrictions evaluation according to age group is observed. In this sense, an important decrease is observed in the number of women with an approved evaluation from the age of 69. In the case of the men, this decrease happens from the age of 74 onwards, confirming the second hypothesis of the study. This decrease in the number of approved individuals, most prominent in the case of women, may be due to a poorer health-related quality of life or women who, despite having a longer life expectancy, have a shorter healthy life expectancy, as evaluated by chronic morbidity and self-perception of health, compared to men [51]. Other factors to bear in mind make references to the practice of driving, which is more frequent in older men than women, with women reducing their amount of driving compared to men independent of the number of kilometers covered prior, their physical health and their cognitive state [52] or for health reasons [42].

The results also indicate an increase in the use of optical corrections with age. It is problems related to visual capacity—mainly surgical aphakia and impaired visual capacity—in relation to psychophysical fitness that are the main cause leading to an evaluation of approved with restrictions, followed by psycho-technical tests—mainly visuomotor coordination, thus fulfilling the third hypothesis of the study. These differences were also found with respect to sex, with men requiring a greater use of optical corrections than women. It should be noted that the use of this type of correction was around 70% in both the group of drivers who were approved and in the group with an evaluation of approved with restrictions. The authors of [53] explain how the factors related to visual ability are those that are mostly related to traffic accidents in older drivers, highlighting visual contrast sensibility, a factor also highlighted by [54]. Specifically, the mesopic vision evaluation tests are the ones that diminish with age, compared to photopic vision tests that are the most suitable for older drivers [55,56].

The number of conditions leading to restrictions increases with age. This confirms the fourth hypothesis of the study, and these are an indicator of poorer performance in driving tasks as noted by [49] when comparing drivers with and without restrictions. In particular, drivers with restrictions are those who show a greater decline in cognitive tasks associated with driving four years after being evaluated for the first time compared to drivers without restrictions [57]. Furthermore, it must be taken into consideration that drivers with the most restrictions are those who have less practice [58,59], which has an impact on driving.

Given the results, it could be asked whether age is a risk factor for driving or, on the contrary, whether the driver’s state of health should be taken into account as a risk factor, which can indeed condition the ability to drive. In the case of older people, certain circumstances occur more frequently than in other age groups. For example, cataracts in older drivers are an age-related pathology. However, even so, the disease is not a risk factor. It would be if the functional deficit it causes or accompanies generated a level of disability that interferes with driving tasks. Thus, in the same example, in the case of cataracts, what is evaluated would be the alteration of the eye’s own functions, such as visual acuity, mesopic vision and contrast sensitivity, because in the case of suffering from lens opacification, these would be the functions that can be altered and interfere with the ability to drive. This factor, the driver’s psychophysical capabilities, is one of the main causes of accident prevention, as stated in Haddon’s model [60].

On the other hand, data from the DGT’s statistical yearbook [30] indicate the incidence of deaths and injuries in people over 65 years of age, with an increase from the age of 75 onwards. These figures may be influenced by other factors, apart from age, such as the greater fragility and vulnerability of the elderly, the type of roads they drive on, the age of the car or the kilometers traveled, because in the latter case, the lack of practice, as driving becomes less frequent with age, may be influencing accident rates in this population group.

From a practical point of view, rather than focusing on the age of drivers, it would be advisable to raise awareness and educate older drivers about the functional deficits they suffer. In this sense, communication campaigns are important to inform and educate users about these aspects [61,62]. These have proven to be effective in promoting the recognition of their own abilities, helping to identify possible deficits presented by users and training or educating them about compensation and adaptation measures that allow them to drive as safely as possible [63,64]. The authors of [57] highlight the importance of formative measures directed at the stimulation of the cognitive functions related to driving, mainly in drivers whose restrictions are associated with said functions.

It is also important to develop specific evaluation protocols for drivers over 65 years of age. The existing ones should be expanded, taking into account decision making and information processing in those road situations in which more resources of this type are required and those which are associated with a higher accident rate among older people, such as intersections, traffic circles, unfamiliar roads or driving under extreme weather conditions [63,64].

### Limitations of the Research and Future Lines of Research

The limitations of the study mainly include the sample size. However, future studies could include samples of older drivers to investigate more than the role of age in the effect of compensatory measures and restrictions on road safety. Future studies should establish the relationship between drivers’ medical conditions, functional deficits and accident risk. Longitudinal studies, which we are already conducting, could provide information on how certain diseases become more frequent with age and how they affect the assessment of the psychophysical fitness of older drivers. On the other hand, the technological evolution of cars is another factor to be taken into account that can further minimize the age variable in driving and even have an impact on driving style and driving skills.

## 5. Conclusions

It can be concluded that, although an increase in the number of cases of restrictions associated with age is observed, the majority of drivers over the age of 65 continue driving. Women are represented to a greater extent in approved evaluations, although it is observed that from the age of 69 onwards, a significant decrease among women is seen in approved evaluations. Furthermore, it is highlighted that visual problems are the main causes for an approved with restrictions evaluation.

The results of the study provide information through scientific evidence on the psychophysical characteristics of drivers over 65 years of age. This information is relevant in establishing the most appropriate preventive measures for safe driving for older drivers, in particular, and indirectly for other drivers and pedestrians who share the roads with them. In view of the results, which indicate significant changes from the age of 70 onwards, especially if the percentage of drivers over this age who have restrictions on the validity of their license is taken into account, bearing in mind the period of validity stipulated at 5 years for drivers over 65 years of age, it would be possible to establish periods of validity of less than these 5 years from the age of 70 onwards. This is covered by European legislation (126/06/EU) [65] which, in its Article 7, states that “Issuance, validity and renewal shall be subject to the following conditions”. “Member States may limit the period of administrative validity set out in paragraph 2 for individual driver’s licenses in any category where it is considered necessary to apply more frequent medical checks or other specific measures such as restrictions for traffic offenders. Member States may reduce the period of administrative validity set out in paragraph 2 for driver’s licenses whose holder’s resident in their territory have reached the age of 50 years in order to increase the frequency of medical checks or other specific measures such as refresher courses. This reduced period of administrative validity may only be applied at the time of renewal of the license”.

## Figures and Tables

**Table 1 healthcare-11-01927-t001:** Distribution of drivers by age group and sex.

Age Group	Total Drivers N (%)	Men N (%)	Women N (%)
65–69 years	597 (35.9%)	386 (64.7%)	211 (35.3%)
70–74 years	627 (37.7%)	510 (81.3%)	117 (18.7%)
75–79 years	439 (26.4%)	392 (89.3%)	47 (10.7%)
Total	1663 (100%)	1288 (77.4%)	375 (22.6%)

Note: N (%) = Number and percentage of drivers participating in the survey.

**Table 2 healthcare-11-01927-t002:** Total number and percentage of drivers over 65 years of age by sex according to the examination result.

Results Recognition	Total Nº (%)	Men Nº (%)	Women Nº (%)	65–69 Nº (%)	70–74 Nº (%)	75 < Nº (%)
Approved	788 (47.5)	557 (43.2)	231 (61.9)	429 (71.8)	301 (48.0)	58 (13.2)
Approved with restriction	830 (49.8)	692 (53.7)	138 (36.5)	164 (27.5)	310 (49.4)	356 (81.1)
Interrupted	27 (1.6)	23 (1.8)	4 (1.1)	3 (0.5)	10 (1.5)	14 (3.1)
Unfit	18 (1.1)	16 (1.3)	2 (0.5)	1 (0.2)	6 (1.1)	11 (2.6)
Total	1663 (100)	1288 (100)	375 (100)	597 (100)	627 (100)	439 (100)

**Table 3 healthcare-11-01927-t003:** Number and percentage of drivers by condition, age group and sex.

	65–69			70–74			75–79			Total
Results Recognition	TD	MD	WD	TD	MD	WD	TD	MD	WD	
Approved	429 54.4%	267 47.9%	162 70.1%	301 38.2%	241 43.3%	60 26.0%	58 7.4%	49 8.8%	9 3.9%	788 100%
Approved with restriction	164 19.8%	118 17.1%	46 33.3%	310 37.3%	256 37.0%	54 39.1%	356 42.9%	318 46.0%	38 27.5%	830 100%
Interrupted	3 11.1%	1 4.3%	2 50.0%	10 37.0%	8 34.8%	2 50.0%	14 51.9%	14 60.9%	0 0.0%	27 100%
Unfit	1 5.9%	0 0.0%	1 50.0%	6 29.4%	5 31.3%	1 50.0%	11 64.7%	11 68.8%	0 0.0%	17 100%
Total	597 35.9%	386 30.0%	211 56.3%	627 37.7%	510 39.6%	117 31.2%	439 26.4%	392 30.4%	47 12.5%	1663 100%

Note: TD (Total Drivers), MD (Men Drivers), WD (Women Drivers).

**Table 4 healthcare-11-01927-t004:** Number and percentage of drivers requiring optical correction.

Evaluation		65–69			70–74			75<			Total
		TD	MD	WD	TD	MD	WD	TD	MD	WD	
Approved	No	143 33.3	87 32.5	56 34.6	90 29.9	75 31.1	15 25.0	15 37.3	10 20.8	5 55.6	248 31.5
	Yes	287 66.7	181 67.5	146 65.4	211 70.1	166 68.9	45 75.0	42 62.7	38 79.2	4 44.4	540 68.5
	Total	430 100	268 100	202 100	301 100	241 100	60 100	57 100	48 100	9 100	788 100
Approved with restriction	No	67 41.1	40 34.2	27 58.7	84 27.1	61 23.8	23 42.6	98 27.5	84 26.3	14 36.8	249 30.0
	Yes	96 58.9	77 65.8	19 41.3	22672.9	195 76.2	31 57.4	259 72.5	235 73.7	24 63.2	581 70.0
	Total	163 100	117 100	46 100	310 100	256 100	54 100	357 100	319 100	38 100	830 100

Note: TD (Total Drivers), MD (Men Drivers), WD (Women Drivers).

**Table 5 healthcare-11-01927-t005:** Diseases (coding by the DGT application and conditions of aptitude of Annex IV of the RGC) presented by drivers over 65 years of age.

Diseases	RC1	RC2	RC3	RC4	Total (%)
Monocular vision	44				44 (3.9%)
Surgical aphakia	302	7			309 (27.8%)
Contrast sensitivity in mesopic vision	1	7	2		10 (<1.0%)
Diplopia	1				1 (<1.0%)
Progressive deterioration of visual ability	182	22	2		206 (18.5%)
Auditory acuity	63	44	1		45 (9.7%)
Partial functional limitation of left leg		1			1 (<1.0%)
Limited cervical mobility	1				1 (<1.0%)
Partial functional limitation of the right leg		1	1		2 (<1.0%)
Total functional limitation of the right leg		2			2 (<1.0%)
Total functional limitation of the left leg	2				2 (<1.0%)
Valvular prosthesis	8	1			9 (<1.0%)
Rhythm disorders	3				3 (<1.0%)
Heart failure	2				2 (<1.0%)
Revascularization surgery < 2 years	7	1			8 (<1.0%)
Revascularization surgery > 2 years	30	12	2		44 (3.9%)
Surgical aneurysm correction	4	3			7 (<1.0%)
Pacemaker	4				4 (<1.0%)
Implantable cardioverter defibrillator	1				1 (<1.0%)
Myocardial infarction	12	5			17 (1.5%)
Chemotherapy treatment	1				1 (<1.0%)
Renal transplantation	5				5 (<1.0%)
Dialysis	1	1			2 (<1.0%)
SAHS	20	12	2		34 (3.1%)
Diabetes	24	48	14	1	87 (7.8%)
Seizures	1				1 (<1.0%)
Transient ischemic attack < 3 years	7	2			9 (<1.0%)
Transient ischemic attack > 3 years	1				1 (<1.0%)
Mood disorder	2				2 (<1.0%)
Movement estimation	37	23	4	1	28 (5.8%)
Visuomotor coordination	58	43	9	4	56 (10.3%)
Other causes	6	3			9 (0.8%)
Total	830 (74.8%)	238 (21.4%)	37 (3.3%)	6 (0.5%)	1111 (100%)

Note: RC (Restrictive Condition); 1–4 (number of conditions restricting driving).

**Table 6 healthcare-11-01927-t006:** List of restrictions imposed on drivers over 65 years of age.

Restrictions	R1	R2	R3	R4	Total
**1-year validity period**	114	2			116 (11.7%)
**2-year term**	292	3	1		296 (29.7%)
**3-year period of validity**	353	7			360 (36.1%)
**Adaptation of rear-view mirrors**	69	83			152 (15.2%)
**Daytime driving limitation**		6	5		11 (1.1%)
**Limitation of driving radius**			3	4	7 (0.7%)
**Speed limitation**		33	7	4	44 (4.4%)
**Automatic gear ratio selection**	2	3			5 (0.5%)
**Left-foot-operated brake pedal**			2		2 (0.2%)
**Hand-operated brake**		1	1		2 (0.2%)
**Hand-operated accelerator**			1		1 (0.1%)
**Accelerator pedal operated by the left foot**				1	1 (0.1%)
**Total**	830 (83.2%)	138 (13.8%)	20 (2.0%)	9 (1.0%)	997 (100%)

**Table 7 healthcare-11-01927-t007:** Psychophysical conditions leading to an unfit or interrupted result by age.

Psychophysical Conditions	65–69 Years	70–74 Years	75<	Total
Progressive disease		1		1 (1.4%)
Binocular visual acuity less than 0.5	1	3	7	11 (16.2%)
Progressive disease when it prevents reaching vision levels	3	3	3	9 (13.2%)
Disease of the central nervous system that prevents driving	1	1	3	5 (7.4%)
Transient ischemic accident that prevents driving		1	2	3 (4,4%)
Balance disturbance			1	1 (1.4%)
Recurrent ischemic stroke		1		1 (1.4%)
Dementia and other cognitive disorders		2	3	5 (7.4%)
Mood disorder		1	1	2 (2.9%)
Movement estimation unfit for driving		5	4	9 (13.4%)
Visuomotor coordination unfit for driving		6	9	15 (22.1%)
Other unspecified causes unfit		2	4	6 (8.8%)
Total	5 (7.4%)	26 (38.2%)	37 (54.4%)	68 (100%)

Number of conditions leading to an evaluation of approved with restrictions.

**Table 8 healthcare-11-01927-t008:** Number and percentage of drivers by age group according to number of conditions.

Nº Conditions	65–69 Years	70–74 Years	75 < Years
1 Condition	163 (90.2%)	310 (74.3%)	357 (63.8%)
2 Conditions	16 (9.2%)	80 (19.2%)	129 (30.9%)
3–4 Conditions	1 (0.6%)	27 (6.5%)	22 (5.3%)
Total	180 (100%)	417 (100%)	508 (100%)

## Data Availability

The data are not public because the study participants were not informed of this.

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
