# Peer review of "Evaluation of Psychophysical Fitness in Drivers over 65 Years of Age"

_healthcare, 2023, doi:10.3390/healthcare11131927_

Round 1

Author Response

The authors of the manuscript are grateful for the comments offered by the reviewer.

Title: Evaluation of psychophysical conditions in drivers over 65 years of age

The manuscript (MS) needs substantial improvements. There are mistakes such as typos,  sentence fragments, and incomplete paragraphs. Please proofread the manuscript (MS), and  rewrite the MS with the help of more professional English writer (if deemed necessary). A number  of improvements can be made and incorporated into the MS. Suggestions for improvements are  as follows:

COMMENT: Title: Do the authors mean 'psychophysical' or 'psychomotor'?

ANSWER: The term 'psychophysical' is used as this is the term used in the Spanish regulations, although in the title the word 'conditions' is changed to 'fitness'. This change is made because the Spanish regulations refer to 'psychophysical fitness'.

COMMENT Abstract:

  1. The term ‘psychophysical’ (or whatever the more appropriate term) should be mentioned.

The objective has been introduced in the objective and the wording has been changed accordingly

  1. A stronger background is suggested (e.g., the effects on road safety rather than that the

subject has been debated in the literature).

This part of the abstract has been modified as suggested by the reviewer by including the following sentence "The deterioration of cognitive and psychophysical abilities associated with ageing has an effect on road safety, in particular on driving".

  1. Please state the objective of the study following the background statement. The authors

should also mention that this study pertained to population in Spain.

Spain has been included in the objective of the study itself.

  1. Objective: Analyze the evaluations? Or evaluate (or examine) the data…?

The assessment of psychophysical skills has been specified.

  1. Health status is also described here. Is it possible to find a term in the title that also

covers this health issue?

The term "health" has been removed from the abstract. In the method section, it shall be specified that it includes the assessment of psychophysical aptitudes according to the Spanish road traffic regulations.

  1. The methods used should be described more clearly.

The method has been described in more detail taking into account the limitations of words included in an abstract.

  1. Please state the conclusions more succinctly.

The paragraph on conclusions has been amended.

COMMENT: Keywords

  1. Please choose keywords that are more relevant to the abstract and content of this

study.

The term road safety has been replaced by traffic.

  1. Check for typo.

The error has been corrected.

COMMENT: Introduction

1.. The authors may want to restructure this section, to make the flow of thought clearer.

Changes have been made to the introduction and new citations have been included.

  1. Suggest citations after the first sentence.

Three citations have been included.

  1. DGT? The meaning of the acronym, Dirección General de Tráfico, has been specified.
  2. Please define and describe 'psychophysical'. Also psychomotor, psychotechnical (used in

the Results Section)

The purpose of assessing psychophysical aptitudes has been specified in the introduction section as well as what is assessed in these tests according to the Spanish driving regulations.

The term psychomotor has been replaced by visuomotor.

In relation to the psychotechnical tests, in the results section it is specified that there are two: estimation of movement and visual-motor coordination.

The article does not describe these two tests as such, although for the reviewer to have an overview of them, the estimation of movement test consists of a ball that is hidden under a rectangle. The ball is thrown several times at different speeds and when the person considers that it has reached the end of the rectangle, depending on the speed at which it appears, the person has to press a button.

The second test, visual-motor coordination, requires the person to operate two levers and drive along the roads that appear on the computer without leaving them.

  1. Suggest the issue of ageing and driving (and roadway safety) to be introduced first,

followed by the issue faced in Spain. Please described the recent findings with regard to

the former issue.

The issue of ageing has been included as the first paragraph, followed by the case of Spain.

  1. Citations are needed for line 47 - 50.

Several citations have been included as requested by the reviewer.

  1. A non complete sentence (line 86).

The error has been corrected.

  1. Suggest a general hypothesis following the objective of the study. Detailed hypotheses

can be described in the Methods Section.

As suggested by the reviewer, a general hypothesis and specific hypotheses have been introduced, although the latter have been included immediately after the general hypothesis and not in the method section.

COMMENT: Methods

  1. Please state clearly whether data collections were performed by the authors (vs. the use

of secondary data).

It has been specified that the data were collected by the researchers themselves.

  1. Please consider moving Table 1 to the Results Section (it seems more appropriate).

It has been preferred to keep the table in the sample description section as the purpose of the table is to describe the sample by condition.

  1. The Procedure Sub-Section can be rewritten better. The authors may want to avoid

using bullets.

As suggested by the reviewer, the use of bullets has been removed.

  1. Provide citation for the choice of statistical analysis.

Added to the corresponding citation.

COMMENT: Results

  1. What is meant by ‘Subsection’? (line 159)

This was an error. The corresponding heading entitled: Number and percentage of drivers by assessment status according to age group and gender has been included.

  1. In describing statistical results, please ensure the correct way in writing the results.

Refer to common statistical books.

It has been corrected as suggested by the reviewer.

  1. ‘Psychotechnical’ was used in this section (line 254). What does this mean? It has not

been described anywhere else.

This includes two tests to be performed by the person who has to renew the driving licence. A coordination test and a movement estimation test. These two tests are mentioned but not described in the article. What each test consists of is specified above.

  1. In general, this section seems difficult to follow. Rewriting this section in a clearer way is

suggested.

Minor changes have been made.

  1. Is it true that most of the results are only describing the classifications and whether the

differences are statistically significant or not?

The results present the frequency and percentage of people for each category. The chi-square is used for the existence of a statistically significant difference between the expected frequency and the observed frequencies in one or more categories in a contingency table.

COMMENT: Discussion

  1. This section seems to be a bit superficial. It is advised that the authors do not merely redescribe the results but, rather, discuss theoretical implications of the results.
  2. It is suggested starting the discussion with major findings and discuss how the findings

compare with previous studies. Please address in more detail the psychophysical and

health aspects, and how these become roadway safety risk.

  1. Degradations in psychophysical capabilities as a function of age will need to be

addressed sufficiently. Changes in cognitive functioning? Psychomotor?

  1. Several of the literatures used have to do with fitness to drive. This concept, however,

was not discussed in this section.

  1. Rewriting the entire section is advised, placing emphasis on (among others) more

theoretical contributions of the findings.

ANSWER: Following the reviewer's suggestions, the discussion has been modified and new citations have been introduced to support the results found.

COMMENT: Conclusions

  1. Similar to the Discussion Section, the authors may want to start with major findings of

their study.

  1. Please state explicitly contributions of the study findings.
  2. The authors may want to state how the findings relate to driving performance and

potential road safety mitigation strategies.

  1. At the end of the section, the authors can suggest recommendations for future studies

ANSWER: The conclusions section has been modified by first highlighting the most relevant conclusions of the study.

Reviewer 2 Report

I would like to thank the authors for their valuable contribution to the literature. The study definitely touches on an important point in the literature on the ageing population and also promises to have a strong impact and points for further practical implications. Below are some comments and recommendations for the manuscript.

1) Although sex differences are important, as can be seen from the data presented, there are some inequalities between female and male drivers. It may be useful to address these in the introductory section, together with the differences between male and female drivers. 

2) As can be seen by the diseases and considering the previous literature, the results could provide a more practical discussion on night driving (Gruber, N., Mosimann, U. P., Müri, R. M., & Nef, T. (2013). Vision and night driving performance of older drivers. Traffic injury prevention, 14(5), 477-485. & Kimlin, J. A., Black, A. A., & Wood, J. M. (2017). Nighttime driving in older adults: effects of glare and association with mesopic visual function. Investigative ophthalmology & visual science, 58(5), 2796-2803.) and driving skills and confidence (Ozturk, I., & Merat, N. (2023). Driving at night and how it's influenced by perceived driving skills. In Contemporary Ergonomics & Human Factors. Taylor & Francis).

3) Please consider organising your results and discussion in the same order as your hypothesis.

4) Table 2) Please also include information on sex by age group.

5) I may have missed that table 3 also shows sex differences. Could you please clarify this? 

6) If the data allows, could you please present a table of diseases by age group? It might be valuable to show how certain diseases (if any) are observed across different age groups.

7) Please either use sex or gender for consistency.

The overall quality of English is nice and well written. I would like to suggest a final check of the punctuation. 

Author Response

The authors would like to thank the reviewer for his comments.

Comments and Suggestions for Authors

I would like to thank the authors for their valuable contribution to the literature. The study definitely touches on an important point in the literature on the ageing population and also promises to have a strong impact and points for further practical implications. Below are some comments and recommendations for the manuscript.

  • COMMENT: Although sex differences are important, as can be seen from the data presented, there are some inequalities between female and male drivers. It may be useful to address these in the introductory section, together with the differences between male and female drivers. 

Aunque las diferencias de sexo son importantes, como se puede ver en los datos presentados, existen algunas desigualdades entre conductores femeninos y masculinos. Puede ser útil abordarlos en la sección introductoria, junto con las diferencias entre conductores masculinos y femeninos.

ANSWER: New references to differences between male and female drivers have been introduced in the introductory section.

  • COMMENT: As can be seen by the diseases and considering the previous literature, the results could provide a more practical discussion on night driving (Gruber, N., Mosimann, U. P., Müri, R. M., & Nef, T. (2013). Vision and night driving performance of older drivers. Traffic injury prevention, 14(5), 477-485. & Kimlin, J. A., Black, A. A., & Wood, J. M. (2017). Nighttime driving in older adults: effects of glare and association with mesopic visual function. Investigative ophthalmology & visual science, 58(5), 2796-2803.) and driving skills and confidence (Ozturk, I., & Merat, N. (2023). Driving at night and how it's influenced by perceived driving skills. In Contemporary Ergonomics & Human Factors. Taylor & Francis).

ANSWER: The importance of assessing mesopic vision has been introduced into the discussion, especially in older groups, considering that it decreases with age and is the vision that exists in low-light conditions, as would be the case for night driving. For this purpose, the citations of Gruber, N., Mosimann, U. P., Müri, R. M., & Nef, T. (2013) and Kimlin, J. A., Black, A. A., & Wood, J. M. (2017), among others, have been included.

  • COMMENT: Please consider organising your results and discussion in the same order as your hypothesis.

ANSWER: The reviewer's commentary presenting the results and discussion according to the hypotheses has been taken into account.

  • COMMENT: Table 2) Please also include information on sex by age group. I may have missed that table 3 also shows sex differences. Could you please clarify this?

ANSWER: All this information is collected in tables 2 and 3. Table 2 specifies frequency and percentage of age and sex separately by condition. Table 3 already specifies for each age group the frequency and percentage of males and females by condition. The main purpose of putting these data in two tables is to make the information easier for the reader.

  • If the data allows, could you please present a table of diseases by age group? It might be valuable to show how certain diseases (if any) are observed across different age groups.

ANSWER: We are currently working on it. In further work we will include the reviewer's suggestion from a longitudinal perspective.

  • COMMENT: Please either use sex or gender for consistency.

ANSWER: As suggested by the reviewer, only one term has been used, sex per gender.

Comments on the Quality of English Language

COMMENT: The overall quality of English is nice and well written. I would like to suggest a final check of the punctuation. 

RESPONSE: The punctuation has been revised again as suggested by the reviewer.

Reviewer 3 Report

The authors investigate the license renewal status and physical condition of older drivers age 65 and older. They suggest that factors other than age should be noted and that communication should be promoted. Although this is an important topic for transportation policy, there are several issues in the manuscript that need to be improved.

Abstract
There seems to be a discrepancy between the content of the manuscript and the content of abstruct such as the conclusions, etc. The structure should be reviewed.

Introduction
Redundant. Please describe what we know so far from previous studies and the rationale for this study, focusing on the points

METHODS
Abbreviations in the table are unclear. Also, the , and . are mixed in the table. Please unify them.

It was stated that a consent form was obtained at the end of the document, but the ethical considerations should also be stated in the text. It should be clearly stated which institution approved the research.

Results
Statistics are described in the main text, but they are not immediately apparent from the tables. It would be easier for readers to understand if the statistical analysis results were also included in the tables. The tables should also be more clear.

There is no explanation of what RC1-4 refers to.

Discussion
A discussion of the results obtained is needed. There are too many tables, and I don't think any statistical analysis has been done on the causal relationship between the results and driving aptitude. Contrast with previous studies is also needed.

I think it would be better to have a discussion from the perspective of transportation policy and the medical aspect.

It's a little confusing to read, so I suggest you brush up on it.

Author Response

The authors of the manuscript are grateful for the comments offered by the reviewer.

Comments and Suggestions for Authors

The authors investigate the license renewal status and physical condition of older drivers age 65 and older. They suggest that factors other than age should be noted and that communication should be promoted. Although this is an important topic for transportation policy, there are several issues in the manuscript that need to be improved.

COMMENT: Abstract
There seems to be a discrepancy between the content of the manuscript and the content of abstruct such as the conclusions, etc. The structure should be reviewed.

ANSWER: The abstract has been modified.

COMMENT: Introduction
Redundant. Please describe what we know so far from previous studies and the rationale for t

ANSWER: The introduction has been modified by including references to psychophysiological functions and gender differences in relation to driving. his study, focusing on the points

COMMENT: METHODS
Abbreviations in the table are unclear. Also, the , and . are mixed in the table. Please unify them. It was stated that a consent form was obtained at the end of the document, but the ethical considerations should also be stated in the text. It should be clearly stated which institution approved the research.

ANSWER: It has been specified what is meant by N(%) in table 1 and the ethics committee has been included.

Results
Statistics are described in the main text, but they are not immediately apparent from the tables. It would be easier for readers to understand if the statistical analysis results were also included in the tables. The tables should also be more clear.

ANSWER: It has been decided to keep the test results in the text and not in the tables, as if they were included in the tables we consider that the tables would have much more information, making it more difficult for the reader.

There is no explanation of what RC1-4 refers to.

ANSWER: The meaning of CR has been specified below the table. The number is due to the fact that a person may have more than one driving restriction. These are recorded as 1 referring to the first condition specified in the assessment, 2 to the second, and so on.

COMMENT: Discussion
A discussion of the results obtained is needed. There are too many tables, and I don't think any statistical analysis has been done on the causal relationship between the results and driving aptitude. Contrast with previous studies is also needed.

I think it would be better to have a discussion from the perspective of transportation policy and the medical aspect.

ANSWER: The discussion has been rewritten and new references related to the results obtained have been included.

Comments on the Quality of English Language

It's a little confusing to read, so I suggest you brush up on it.

ANSWER: Editing and translation have been revised.

Round 2

Reviewer 1 Report

Please double check for long sentences, which sometimes are difficult to comprehend

Author Response

COMMENT: Please double check for long sentences, which sometimes are difficult to comprehend

ANSWER: In some paragraphs, sentences have been made shorter as suggested by the reviewer.

Reviewer 2 Report

Thank you for the review. Below are some minor comments that need to be addressed before the study can be considered for publication.

1) Please be consistent with the use of men-women or male-female. Do not mix and match.

2) Please include in the manuscript any follow-up plans that you mentioned in the response letter.

N/A

Author Response

COMMENT:  Please be consistent with the use of men-women or male-female. Do not mix and match.

ANSWER: It has been corrected. It has been decided to use the terms men and women.

COMMENT:  Please include in the manuscript any follow-up plans that you mentioned in the response letter.

ANSWER: The following sentence has been included in the section on future studies “Longitudinal studies, which we are already conducting, could provide information on how certain diseases become more frequent with age and how they affect the assessment of the psychophysical fitness of older drivers.

Reviewer 3 Report

Still not corrected [,] in Tables 1 and 4.

Table 3 needs explanation of C, CH, and CM.

Author Response

COMMENT: Still not corrected [,] in Tables 1 and 4.

ANSWER: The error has been corrected as suggested by the reviewer.

COMMENT: Table 3 needs explanation of C, CH, and CM.

ANSWER: The meaning of initials has been specified. C has been replaced by TD (Total drivers), CH by MD (Men drivers) and CM by WD (Women drivers).